

# Leveraging Machine Learning to Enhance Aerosol Classification using Single-Particle Mass Spectrometry

Jose A. Perez Chavez[1], Maria Zawadowicz[2], Christopher Boxe[1,3], Joseph Wilkins[1,3]

[1]Howard University Program for Atmospheric Science (HUPAS), Howard University, Washington, D.C. 20059, USA
[2]Environmental and Climate Sciences Department, Brookhaven National Laboratory, Upton, NY 11973, USA
[3]Department of Earth, Environment and Equity, Howard University, Washington, D.C. 20059, USA

*Correspondence to*: Jose Perez Chavez (jose.perezchavez@bison.howard.edu)

**Abstract.** Advancing automated classification of atmospheric aerosols from Single-Particle Mass Spectrometry (SPMS) data
remains challenging due to overlapping chemical signatures and limited labeled data. Semi-supervised learning approaches offer potential solutions by leveraging unlabeled data to enhance classification accuracy. Four models were compared: a supervised Support Vector Machine (SVM), a self-training SVM, a stacked autoencoder classifier, and a stacked autoencoder trained with a temporal ensembling mean teacher framework. All models achieved robust performance with overall accuracies of 90.0-91.1%, representing improvements over previous work on the same dataset (87%) and competitive
performance with current methods. Notably, the models effectively classified aerosols with limited representation in the dataset—soot (0.77% of spectra, F1-scores: 0.93-0.97) and hazelnut pollen (0.98% of spectra, F1-scores: 0.97-1.00)—highlighting their ability to capture distinct chemical signatures even with fewer than 200 training samples per class. While challenges persist in classifying certain species, particularly feldspars due to overlapping spectral features and class imbalances, this study demonstrates the significant potential of semi-supervised learning and advanced machine learning
architectures in improving aerosol classification, with implications for atmospheric and climate research.

## 1 Introduction

Understanding the composition of atmospheric aerosols is essential for evaluating their impact on climate, environment, and human health (Seinfeld and Pandis, 2016). Aerosols, tiny solid or liquid particles suspended in the air, have a significant and complex impacts on various environmental aspects. They influence Earth's radiation balance by scattering (Atwood et al.,
2019) and absorbing visible (Ravishankara et al., 2015) and infrared radiation, leading to potential cooling or warming effects depending on the aerosol type and location (Bellouin et al., 2005; Myhre et al., 2014). Aerosols act as cloud condensation nuclei (CCN), and ice-nucleating particles (INPs), the seeds around which water vapor condenses to form clouds (DeMott et al., 2010, 2016; Farmer et al., 2015; Lohmann and Feichter, 2005). This interaction influences cloud properties like reflectivity, lifespan, and precipitation patterns, with implications for regional weather and water cycles
(Andreae et al., 2004; Andreae and Rosenfeld, 2008). Additionally, high concentrations of certain aerosols, such as fine



particulate matter, pose a significant threat to human health, increasing the risk of respiratory and cardiovascular diseases (Fuller et al., 2022; Pope and Dockery, 2006). Despite their importance, accurately measuring and characterizing aerosols, and their intricate atmospheric interactions remains a challenge (Christopoulos et al., 2018; Riemer et al., 2019). This hinders climate models' ability to precisely quantify direct and indirect radiative forcing and cloud processes (Boucher et al., 2014),

leaving a degree of uncertainty in our understanding of their full climatic effects.

Single-particle mass spectrometry (SPMS) offers real-time analysis of individual aerosol particles' chemical composition (Murphy, 2007). SPMS utilizes a pulsed UV laser for the ablation and ionization of single aerosol particles and then accelerates the ions into a time-of-flight mass spectrometer to determine chemical composition (Cziczo et al., 2006; Gard et al., 1997; Pratt et al., 2009; Su et al., 2004). SPMS permits the analysis of aerosol particles in the size range of ~150-3000

nm in diameter (Li et al., 2011; Murphy, 2007; Zawadowicz et al., 2017). SPMS can provide information on the size, shape, elemental composition, and mixing state of aerosol particles, which are important parameters for determining their sources and impacts (Cziczo et al., 2004). The mass spectra obtained from SPMS can be used to identify the source of the particles, such as biomass burning, based on specific chemical markers, like the presence of an ion peak at m/z 213 ($^{39}K_3{}^{32}SO_4{}^+$) (Silva et al., 1999). The chemical composition of the particles can be linked to their impact, for example, automobile exhaust

particles containing hydrocarbons, Al+, Pt+, and Pb+ (Noble and Prather, 1996). Similarly, some work has found that some ice residuals exhibited characteristic mass spectral features of isoprene-derived organosulfates, specifically at mass-to-charge (m/z) ratios of 211, 213, and 215 (Wolf et al., 2020). These peaks indicate the presence of isoprene-epoxydiol-derived organosulfates (IEPOX-OSs) in INPs. SPMS can also differentiate between different types of particles, for instance, distinguishing between internally and externally mixed particulate populations (Song et al., 1999).

Identifying aerosols from complex SPMS data is difficult because individual particles contain millions of molecules with hundreds of distinct chemical species (Riemer et al., 2019), both primary (like elemental carbon and metals from combustion) and secondary (such as nitrates from NOx oxidation and sulfates from SO2 oxidation) (Riemer et al., 2019). Atmospheric processing further complicates classification as particles undergo coating with secondary materials (e.g., soot becoming coated with sulfate), coagulation, and cloud processing, leading to mixed compositions where aerosol origins and

atmospheric processing create overlapping chemical signatures (Ault and Axson, 2017; Murphy and Thomson, 1997; Prather et al., 2008), with even minor compositional changes significantly altering their climate impact (Riemer et al., 2019; Usher et al., 2003). Mass spectral analysis faces specific technical challenges: organic compounds fragment during laser ablation producing soot-like carbon progressions (Zelenyuk et al., 2008), alkali metals from urban dust and biomass burning dominate spectra and mask other components (Zelenyuk et al., 2008) due to their high ionization efficiency, and the laser

ablation process itself shows variable sensitivity to different components while producing inconsistent spectra from identical particles (Phares et al., 2001). These complications make it particularly difficult to determine exact mass fractions at the single-particle level - for instance, fly ash, mineral dust, and bioaerosol can all exhibit strong phosphate signals in their spectra (Christopoulos et al., 2018). Even distinguishing bioaerosols from nonbiological phosphorus-rich particles, crucial for understanding ice nucleation, is challenging (Zawadowicz et al., 2017). This difficulty stems from overlapping ion



markers where phosphate (PO⁻, PO₂⁻, PO₃⁻) and organic nitrogen ions (CN⁻, CNO⁻) traditionally used as bioaerosol indicators also appear in nonbiological particles such as vehicular exhaust, mineral dust, and fly ash.

Early approaches to the classification of aerosols relied on user-defined rules to determine the category of a particle. These rules were initially developed for computer-controlled scanning electron microscopy (CCSEM) coupled with energy-dispersive X-ray spectroscopy (EDX) (Casuccio et al., 1983; Kim et al., 1987). These datasets typically contained information on hundreds to thousands of particles and approximately 15-30 elements, making manual spot checks feasible (Riemer et al., 2019). However, the advent of field-deployable real-time single-particle mass spectrometry (SPMS) (Hinz et al., 1994; McKeown et al., 1991; Prather et al., 1994) led to a significant increase in both data complexity and volume. This rendered traditional classification methods impractical, driving the development of more advanced techniques.

Instruments like Aerosol Time-of-Flight Mass Spectrometry (ATOFMS), laser ablation aerosol particle time of flight mass spectrometer (LAAPToF), and Particle Analysis by Laser Mass Spectrometry (PALMS) (Cziczo et al., 2006; Pratt et al., 2009; Zawadowicz et al., 2020), present a data analysis bottleneck, as they can characterize thousands of particles per minute – a rate exceeding that required to manually analyze the data (Song et al., 1999). Unsupervised learning models have become essential tools for handling such large datasets. Among these, Adaptive Resonance Theory 2a (ART-2a) has been widely used to cluster particles based on their chemical composition (Song et al., 1999). The algorithm identifies unique clusters, which are then labeled for interpretation, offering flexibility and control over the categorization process. This method is very popular and it is still being used in the last decades (Axson et al., 2016). For example, they were used to isolate and assign Fresh vs Aged Soot membership of aerosol samples from Mexico City and Riverside, CA (Moffet and Prather, 2009).

Another popular clustering algorithm is KMeans, which groups data by minimizing within-cluster variance (Anderson et al., 2005; Gross et al., 2010). KMeans is known for scalability but requires specifying the number of clusters beforehand. Atwood et al. (2019) classified aerosol population types and cloud condensation nuclei properties representative of the coast and interior of California using KMeans cluster analysis. Both ART-2a and KMeans are unsupervised techniques, meaning they don't need predefined labels, making them valuable for the often difficult-to-label aerosols. Even for studies where aerosol identification takes a more empirical and rigorous approach such as evaluating the SPMS against reference spectra measured at a lab, checking for signal at specific markers, analyzing relative peak intensities, clustering takes role in defining classes of aerosols (Cziczo et al., 2004; Freutel et al., 2013; Froyd et al., 2019; Shen et al., 2019). Despite their advantages, these unsupervised algorithms have limitations. Parameters require careful tuning and assessing cluster validity is crucial (Song et al., 1999). Additionally, assumptions of cluster shape and the impact of high-dimensional data can pose challenges. Researchers have explored incorporating expert knowledge to refine clustering results (Zelenyuk et al., 2008), demonstrating the ongoing efforts to improve the accuracy and robustness of single-particle analysis methods.

Supervised classification algorithms offer several advantages over unsupervised clustering algorithms (Beck et al., 2024). They learn from labeled data, tailoring classifications to specific goals, potentially increasing generalizability across similar applications. They often rely on optimized methods, reducing the need for manual calibration. Some can handle non-linear relationships between features and labels, leading to more accurate classifications (Gong et al., 2022). They may employ



normalized metrics or reduction of dimensions to address poor model generalization and stability in high-dimensional
spaces, and are optimized for making predictions, potentially achieving higher accuracy (Beck et al., 2024).

Previous studies have shown the potential of supervised learning in classification and feature extraction of single-particle
mass spectra for aerosols. Support Vector Machine (SVM), a supervised machine learning algorithm (Cortes and Vapnik,
1995) which could find a clear gap between separate labeled categories, has been successfully applied in binary classification
for differentiate bioaerosol mass spectra from phosphorus-bearing spectra achieving 97% overall accuracy (Wang et al.,
2024c; Zawadowicz et al., 2017). At the same time this model has been used to determine representative SPMS signatures of
three aerosol types for apportionment of coal combustion sources (Xu et al., 2018). Another popular model, the Random
Forest classifier, has shown promising results. The RF has previously been applied to the same dataset used in this paper, in
classifying 20-unique classes of aerosols, achieving overall accuracy of 87% and identifying important influential markers
(Christopoulos et al., 2018). Another study demonstrated that a random forest machine learning algorithm can effectively
classify single particles based on single-particle soot photometer data, achieving a classification accuracy of ≥99% for rBC
and FeOx, and 47%–66% for other aerosol types, while broader categories demonstrated high accuracies (Lamb, 2019).

In a separate study, supervised learning algorithms were compared in classifying eight distinct broad categories
representative of common aerosol types found in Central Europe during summer (Wang et al., 2024c). All algorithms
achieved overall accuracies above 94%, with RF, SVM, and Multi-Layer Perceptron (MLP) leading the way with accuracies
exceeding 97% (Wang et al., 2024c). The work indicates that these supervised algorithms offered significant advantages
over unsupervised clustering methods, primarily due to their ability to learn from labeled data, tailor classifications to
specific research goals, and potentially achieve higher accuracy in high-dimensional SPMS datasets (Wang et al., 2024c).
However, a fundamental limitation of supervised learning is that these algorithms cannot identify aerosol types absent from
the training data. This presents a practical challenge for atmospheric applications where novel or unexpected particle types
may be encountered during field campaigns. Unlike unsupervised clustering methods that can potentially identify new
particle groups, supervised models will forcibly classify unknown particles into one of the predefined categories from the
training set, potentially leading to mischaracterization of atmospheric composition when deployed in diverse environments.

More recent work has explored the application of deep learning, particularly convolutional neural networks (CNNs), for the
classification of aerosol particles. The work demonstrates that both 1D-CNN and 2D-CNN architectures could effectively
classify SPMS data, achieving accuracy exceeding 90% with a sufficiently large labeled dataset (Wang et al., 2023, 2024b,
a). The work highlighted the advantages of CNNs over traditional machine learning methods like multilayer perceptrons,
emphasizing the CNNs' ability to automatically extract meaningful features from the data without the need for expert feature
engineering. Nevertheless, noise augmentation, square root transformation, averaging mass spectra, and swapping positive
and negative mass peaks, have been found to improve classification accuracy from approximately 75% to 86.8% when
training with only 1/8 of the original labeled data (Wang et al., 2024d). Data augmentation is therefore deemed a crucial step
to overcome the challenges of supervised learning such as the high cost and time-consuming process of obtaining accurate



labels (Wang et al., 2024d), potential biases inherited from training data, and the risk of overfitting with limited datasets (Beck et al., 2024).

Semi-supervised learning offers an alternative solution to reducing the over reliance on labeled data by incorporating unlabeled data in the model training. This method is particularly beneficial when labels are expensive to obtain but unlabeled data is abundant (Zhou, 2021). By leveraging unlabeled data, semi-supervised learning can enhance the model's generalization performance and mitigate the risk of overfitting. This study demonstrates increased accuracy in aerosol classification by incorporating unlabeled data through supervised support vector machine (SVM) algorithms and a stacked autoencoder classifier, focusing on two notable training methods: self-training and temporal ensemble mean teacher, which will be explained in the following section.

## 2 Data and Methods

The dataset analyzed in this study was collected using the Particle Analysis by Laser Mass Spectrometry (PALMS) instrument during the Fifth Ice Nucleation workshop (FIN-1) (Shen et al., 2024) at the Karlsruhe Institute of Technology's (KIT) Aerosol Interactions and Dynamics in the Atmosphere (AIDA) facility, with additional samples acquired at MIT's Aerosol and Cloud Laboratory (Christopoulos et al., 2018). The labeled portion consists of 18,827 single-particle mass spectra representing 20 distinct aerosol types, including mineral dust, biological particles, combustion products, and secondary organic aerosol (SOA). Each spectrum contains 193 mass-to-charge (m/z) peaks ranging from 1 to 207 m/z, along with particle time-of-flight as a proxy for vacuum aerodynamic diameter. The version of the PALMS instrument used here is unipolar—capable of acquisition of either positive or negative mass spectra at one time. This study focuses primarily on negative-polarity spectra, which was demonstrated to be effective in discriminating between bioaerosol and dust particles (Zawadowicz et al., 2017).

The most prevalent classes are fly ash (21.19%, 3,957 spectra), K-feldspar coated with SOA (7.91%, 1,478 spectra), and uncoated K-feldspar (7.88%, 1,471 spectra). Several important atmospheric aerosol types have limited representation in the dataset, including soot (0.77%, 143 spectra) and hazelnut pollen (0.98%, 183 spectra). Agar growth medium (1.29%, 240 spectra), while not an atmospheric aerosol type, is included as a potential laboratory contaminant for bacterial particles cultivated on this medium. This class imbalance reflects typical challenges in atmospheric particle collection and presents an opportunity to evaluate model performance with limited training data.

The dataset also includes 14,478 unlabeled mass spectra collected during the same experimental periods. These spectra follow identical formatting and preprocessing steps as the labeled data but lack definitive aerosol type assignments. This unlabeled portion serves as additional training data for our semi-supervised learning approaches.

Data preparation followed a consistent protocol for all experiments. The complete dataset was first randomly shuffled, with 10% reserved for testing. For the labeled portion, we employed stratified sampling to maintain class distributions between training and test sets, crucial given the significant class imbalances. Initial preprocessing included handling missing





diameter-size feature values by setting them to zero if missing, as well as dropping mass spectra where there were only a
single or less mass peaks present. Feature scaling was then performed using a maximum absolute scaler, fitted to the training
data and applied to both training and test sets. This scaling approach was chosen over standard normalization or L1/L2
normalization due to its ability to preserve sparsity in the mass spectra while constraining all features to the range [0,1].

The decision to use a 10% test set was validated through stability analysis comparing performance metrics across test set
sizes of 10%, 20%, and 25%. While individual class metrics showed expected variations with different sampling, overall
performance measures (accuracy, precision, recall, F1-score) remained stable. This suggests the chosen split provides
reliable evaluation of model performance while maximizing available training data. However, metrics for classes with very
limited samples (particularly soot, hazelnut pollen, and agar) should be interpreted with appropriate caution due to their
small test set representations.

## 2.1 Classification Methods

To evaluate the potential of semi-supervised learning for aerosol classification, we implemented and compared four distinct
approaches. Each successive model builds upon traditional supervised learning by incorporating unlabeled data in
increasingly sophisticated ways. All models were implemented in Python using PyTorch for neural network architectures
(Ansel et al., 2024) and scikit-learn for traditional machine learning algorithms (Pedregosa et al., 2011). The following
sections detail the theoretical background, implementation, and training methodology for each classifier.

## 2.2 Support Vector Machine Classifier

Support Vector Machines (SVMs) serve as our baseline supervised classification method, chosen for their documented
success in SPMS data analysis (Christopoulos et al., 2018; Wang et al., 2024c; Zawadowicz et al., 2017) and robust
performance in high-dimensional feature spaces. SVMs operate by mapping input vectors into a high-dimensional feature
space where they construct optimal hyperplanes to separate different classes (Beck et al., 2024; Cortes and Vapnik, 1995).
For our multi-class problem, we employed scikit-learn's Support Vector Classification with a one-vs-one scheme, where
separate binary classifiers are trained for each pair of classes.

Model optimization utilized scikit-learn's HalvingGridSearchCV with cross-validation to efficiently explore the parameter
space. The search focused on three key parameters: kernel type, regularization parameter (C), and kernel coefficient
(gamma). Starting with an initial resource allocation of 2,000 training samples, the algorithm evaluated 30 randomly
sampled parameter combinations, progressively eliminating weaker candidates by a factor of 2 until identifying the optimal
configuration. This approach yielded optimal values of C=284.73 and gamma=0.2412 using a radial basis function (RBF)
kernel, which implicitly maps data into a higher-dimensional space where classes become more linearly separable.

During optimization, we found that maximum absolute value scaling of input features produced consistently better results
compared to standard scaling or L1/L2 normalization. This superiority likely stems from the technique's preservation of
sparsity in the mass spectra, an important characteristic of SPMS data where many m/z channels typically show zero




intensity. The final model maintains high sparsity while achieving effective separation between aerosol classes through the RBF kernel's nonlinear mapping.

## 2.3 Self-Training SVM Classifier

Building upon the baseline SVM, we implemented a self-training approach to leverage unlabeled data through scikit-learn's
SelfTrainingClassifier. This semi-supervised method iteratively expands the training set by incorporating unlabeled samples that the model classifies with high confidence (Yarowsky, 1995). The base classifier remains the optimized SVM described in the previous section, maintaining the same preprocessing pipeline with maximum absolute scaling and missing value imputation.

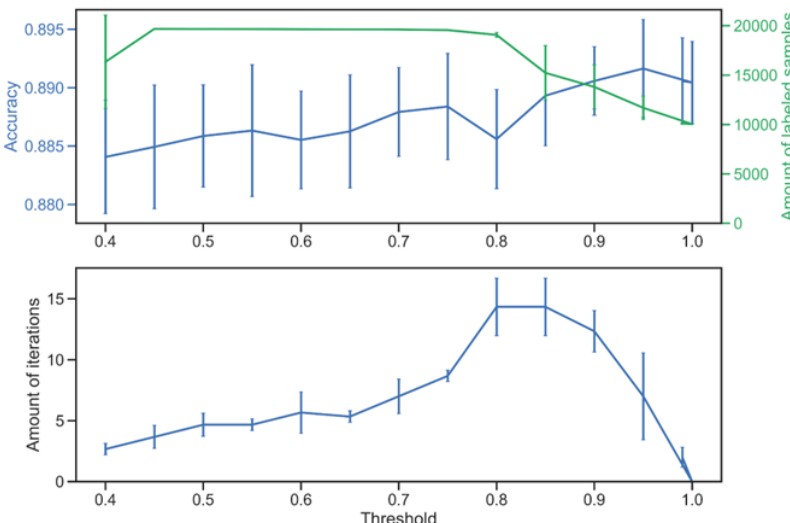


**Figure 1: Effects of threshold hyper-parameter. Top: Cross-validated accuracy (3-fold) as a function of threshold, right axis indicates the number of samples used in training (labeled). Bottom: Number of iterations required to label all possible unlabeled samples.**

A critical parameter in self-training is the confidence threshold that determines which predictions on unlabeled data are
sufficiently reliable to include in subsequent training iterations. We conducted a systematic evaluation of threshold values between 0.5 and 1.0, analyzing their impact on both classification performance and the proportion of unlabeled data utilized (Figure 1). The analysis revealed that threshold values above 0.8 consistently improved classification accuracy, with optimal performance achieved at 0.95. This high threshold results in conservative selection of unlabeled samples, with only approximately 25% of the unlabeled data incorporated into the training set but ensures high reliability of the pseudo-labels.

We explored potential improvements through probability calibration using CalibratedClassifierCV, as accurate probability estimates are crucial for reliable self-training. However, calibration provided negligible improvements in accuracy, precision, recall, and F1-score metrics, suggesting the SVM's decision boundaries were already well-optimized from the initial



supervised training. This robustness to calibration, combined with the limited utilization of unlabeled data even at moderate thresholds, indicates potential limitations in the self-training approach for our specific classification task.

**220  2.4 Stacked Autoencoder Classifier**

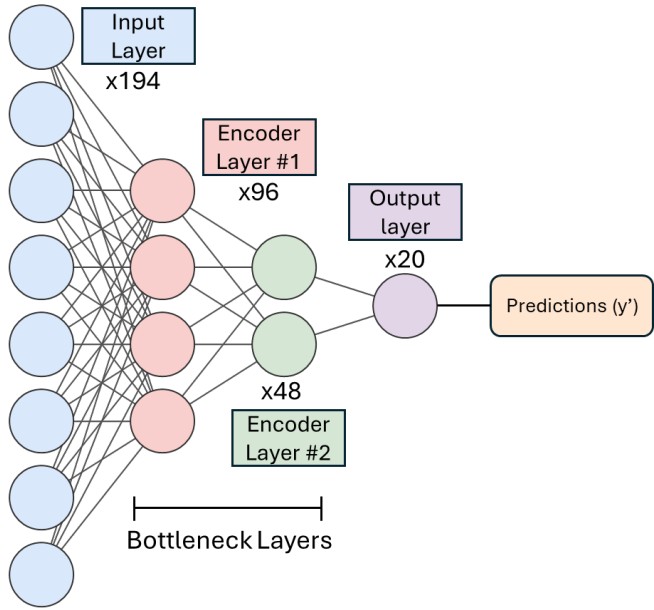

**Figure 2. A qualitative outlook to the Stacked Autoencoder Classifier model, which employs fully connected layers structure, with each layer designated by its size. The model uses a softmax function to activate its predictions, effectively transforming the model's output into probabilities associated with each label.**

The stacked autoencoder represents a more sophisticated approach to aerosol classification by combining dimensionality reduction with supervised learning in a deep neural network architecture. The model consists of two key components: an autoencoder network that learns compressed representations of mass spectra, and a classification layer that operates on these learned features. The complete architecture, illustrated in Figure 2, processes the 194-dimensional input (193 mass peaks + particle time-of-flight) through a series of fully connected layers that progressively reduce dimensionality while preserving
essential spectral characteristics.

The autoencoder training proceeds in three sequential stages. First, an initial autoencoder compresses the 194 input features to 96 latent dimensions, trained using both labeled and unlabeled data to learn general spectral patterns. This pre-trained network is then integrated into a deeper architecture where a second encoder further reduces the dimensionality to 48 features. Both autoencoders employ ReLU activation functions and are trained using the Adam optimizer with rectified
adaptation (Liu et al., 2019) to minimize mean squared error (MSE) loss, with the first stage running for 300 epochs and the second for 100 epochs using a batch size of 12. This staged training approach allows the network to learn increasingly abstract representations of the mass spectra.





The final classification layer transforms the 48-dimensional latent representation into predictions across the 20 aerosol classes using a softmax activation function. This layer is trained exclusively on labeled data using the Adam optimizer with decoupled weight decay regularization (Loshchilov and Hutter, 2017) and cross-entropy loss. The softmax output provides probability distributions over possible aerosol types, allowing for both definitive classifications and uncertainty quantification. Importantly, the autoencoder's ability to capture essential spectral features in its latent space potentially enables more robust classification than direct supervised learning on the raw spectra.

The complete training process optimizes both reconstruction accuracy and classification performance. The reconstruction objective ensures the latent space retains meaningful chemical information, while the classification objective fine-tunes these features for optimal class separation. This dual optimization distinguishes the stacked autoencoder from traditional supervised approaches, potentially offering better generalization when dealing with complex, high-dimensional mass spectra.

## 2.5 Mean Teacher Framework

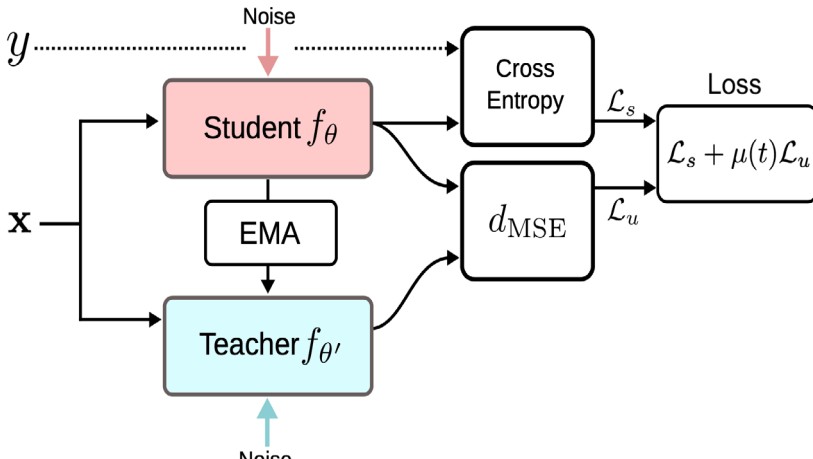

**Figure 3. Overview of the Mean Teacher framework. (Image source: Tarvaninen & Valpola, 2017)**

The Mean Teacher framework extends our stacked autoencoder approach by implementing a consistency-based semi-supervised learning strategy (Tarvainen and Valpola, 2017). This approach maintains two models: a student model that learns actively through standard optimization, and a teacher model whose weights are updated as an exponential moving average of the student model's weights. The framework aims to improve model robustness and generalization by enforcing consistent predictions across different views of the same data.

Building upon the previously described stacked autoencoder architecture, we initialize the student model using the pre-trained encoder-decoder structure. The teacher model begins as an exact copy of the student but evolves differently during training. While the student model updates through standard backpropagation, the teacher model's parameters ($\theta'_t$) are updated at each training step t using an exponential moving average:

$$\theta'_t = \alpha\theta'_{t-1} + (1-\alpha)\theta_t$$



where $\theta_t$ represents the student model's parameters and $\alpha$ is a smoothing coefficient set to 0.999. This averaging process creates a more stable model that typically yields better predictions than the student model alone.

The training objective combines two loss terms: a supervised cross-entropy loss ($L_{ce}$) for labeled data and a consistency loss ($L_{con}$) that encourages agreement between student and teacher predictions on unlabeled data:

$$L_{total} = L_{ce} + \lambda L_{con}$$

where $\lambda$ is a weighting parameter that balances the two objectives. The consistency loss is computed as the mean squared error between the student and teacher predictions on the same input processed with different random augmentations. Both models employ the AdamW optimizer, with the encoders and classification layer trained end-to-end using this combined objective.

The Mean Teacher framework introduces minimal computational overhead during inference since only the teacher model is used for final predictions. However, its primary advantage lies in leveraging unlabeled data to improve model robustness without relying on potentially error-prone pseudo-labeling, as in the self-training approach. This makes it particularly suitable for SPMS data analysis, where obtaining labeled data is expensive but unlabeled spectra are abundant.

Temporal ensembling mean teacher is a more sophisticated method, using consistency regularization for robustness
(Tarvainen and Valpola, 2017). It maintains 'student' and 'teacher' models, constantly updating the student during learning while the teacher serves as a moving average. The student model's predictions on unlabeled data are regularized by minimizing their distance from the teacher model's predictions. This technique enhances the model's robustness and generalization capabilities. This method has yet to be used in any mass spectral data problem.

## 3 Results

This section presents an analysis of the models' performance, focusing on the impact of unlabeled data on classification accuracy and the identification of key features driving aerosol classification.

### 3.1 Reconstruction Performance

The ability of the stacked autoencoder to reconstruct a mass spectrum from a latent compressed representation shows robust performance, with approximately 80% of test samples showing a sum of squares error (SSE) below 0.09. Figure 4 illustrates
this performance using a Moroccan aerosol sample, where the reconstructed spectrum exhibits reconstruction of high-intensity peaks while maintaining the sparsity of zero or low-intensity peaks.





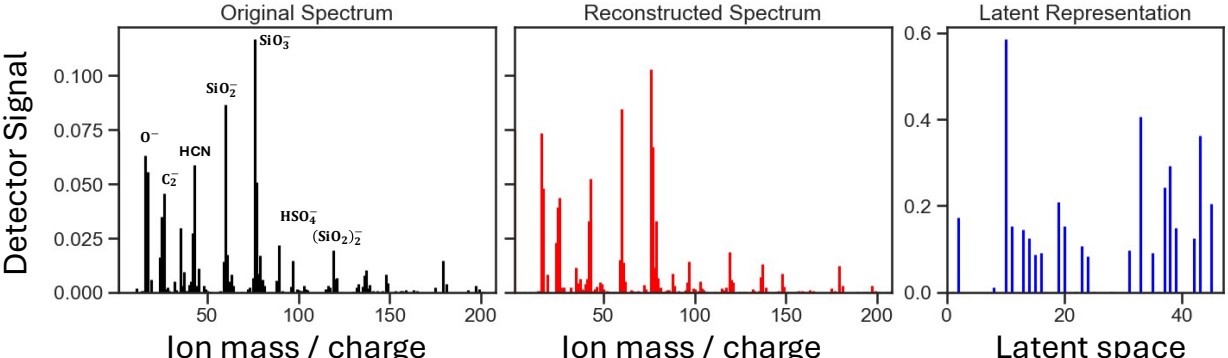

**Figure 4. A Moroccan aerosol mass spectra sample (left), its reconstructed spectra by the stacked autoencoder (middle) and the compressed vector in latent space (right). The reconstruction sum of squares error is 0.09. The y-axis are arbitrary normalized intensities.**

The stacked autoencoder showed consistent reconstruction performance across classes, with Chinese soil, Hazelnut particles, and Argentinian soil having the three lowest (best) average reconstruction errors at 0.015, 0.027, and 0.031, respectively. The average reconstruction error per class was 0.082. Conversely, Cellulose, Soot, and SOA exhibited the three highest (worst) reconstruction errors, indicating less accurate reconstruction.

Reconstruction quality shows no correlation with classification accuracy. Despite their high reconstruction errors, classes like Soot achieved high precision and recall scores, while Chinese and Argentinian samples with very low reconstruction errors showed sub-par classification performance. The distribution of SSE scores per class is included in the appendix.

## 3.2 Overall Model Performance

The metrics used to evaluate classifier performance include:

- Overall accuracy (OA): The proportion of correct predictions
- Precision: The proportion of positive predictions that are correct
- Recall: The proportion of actual positive cases correctly identified
- F1 score: Considers both precision and recall equally for each class
- Confusion matrix (Fig. 6): Depicts the correct identifications and misclassifications

Permutation feature importance analysis is employed to understand model decision-making. This involves shuffling feature values randomly and measuring the resulting decrease in F1 score. Comparing feature importance across models and classes reveals the most influential mass spectral features for each aerosol type. Furthermore, the area under the precision-recall curve (AUC-PR) aids in assessing how well models distinguish between class pairs, especially useful for imbalanced classes. Class-specific precision, recall and F1-scores are available in the appendix in Table A2.

| Model | OA | Precision | Recall | F1 Score |
|---|---|---|---|---|





| | | | | |
|---|---|---|---|---|
| SVM Classifier, #1 | 0.903 | 0.915 | 0.911 | 0.912 |
| Self-Learning SVM, #2 | 0.900 | 0.912 | 0.905 | 0.908 |
| Regular Stacked Autoencoder classifier, #3 | **0.911** | **0.918** | 0.915 | **0.916** |
| Mean Teacher Stacked Autoencoder classifier, #4 | 0.906 | 0.913 | **0.916** | 0.914 |

310 **Table 2. Performance metrics of each model in this study. Left to right: Overall accuracy (OA), Precision, Recall, and F1-Score.**

The models achieved overall accuracies of 90.0-91.1%, representing robust performance for a challenging 20-class classification task characterized by severe class imbalance (e.g., Soot and Hazelnut as rare classes) and high spectral overlap between aerosol types (e.g., Feldspar species). This performance surpasses the 87% accuracy previously reported on this dataset (Christopoulos et al., 2018) and is competitive with the 94-97% accuracies achieved on simpler, 8-class aerosol

315 problems (Wang et al., 2024c). The performance of all classifier models is listed in Table 2 and shown in Figure 5.

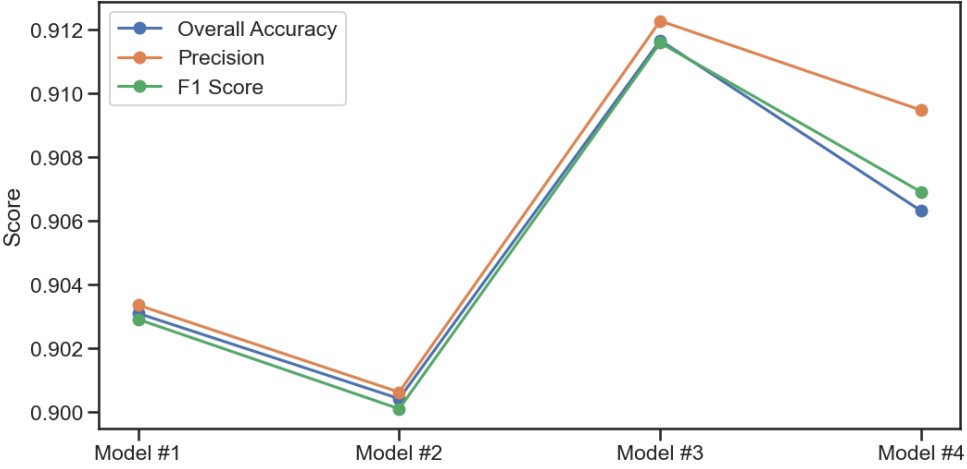

**Figure 5. Overall accuracy, precision and recall for each model in the paper.**

The stacked autoencoder classifier (Model 3) performed best at 91.1% overall accuracy, 91.8% precision and 91.6% F1-score. Model 1 achieved an F1 score of 91.2% but during parameter optimization most iterations yielded a score below

320 89.5%. The performance of each model in each class can be seen in the confusion matrix in Fig. 6. Including the unlabeled data in a semi-supervised fashion for the SVM model showed a slight decrease in performance (Model 2), despite the increase in overall training data. The calibration procedure for the self-learning SVM, as discussed in Sect. 2, demonstrated difficulty in using most of the unlabeled data to increase classification accuracy.



**Figure 6. Confusion matrices. a) standard SVM classifier, b) Self-Training SVM classifier, c) stacked autoencoder classifier, and d) Mean teacher trained stacked autoencoder classifier.**

The small differences in overall classification performance mask improvements in the classification of rare but atmospherically significant aerosol types as seen in Fig. 7. Notably, the stacked autoencoder classifiers (Models 3 and 4) showed better performance for underrepresented classes like Soot, Agar, and Moroccan aerosol types. For example, Model 3 improved Soot classification from an F1-score of 0.93 to 0.97. These class-specific differences become particularly important when examining challenging classification problems where spectral overlap and class imbalances create systematic difficulties across all approaches.




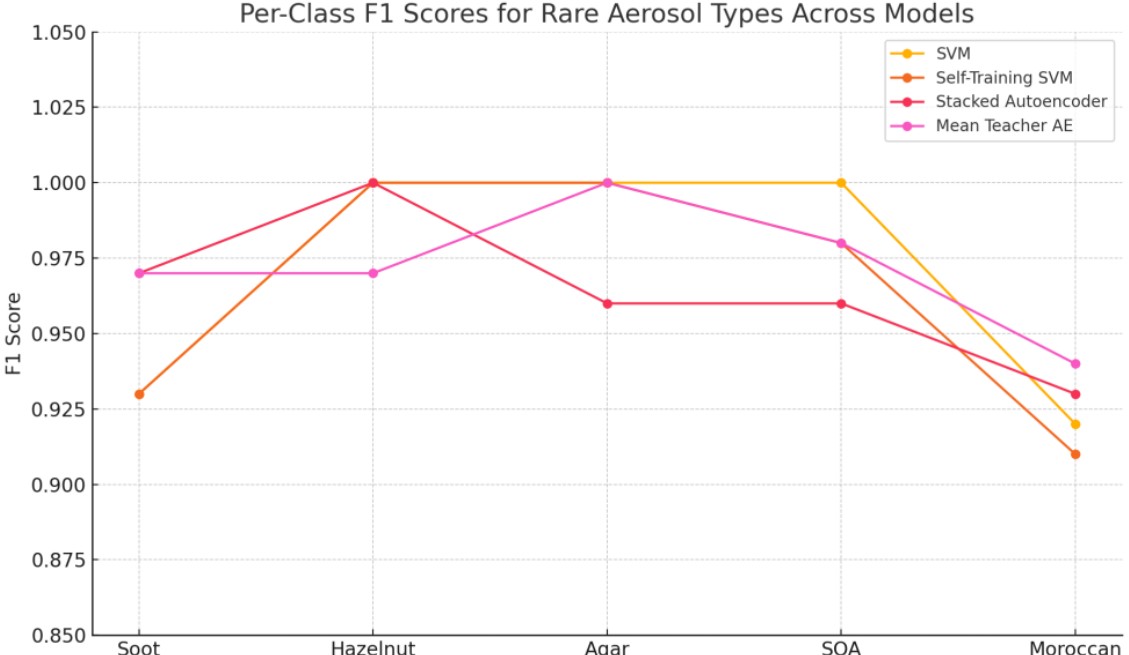

**Figure 7. Comparison of per-class F1-scores across four models for rare aerosol types (Soot, Hazelnut, Agar, SOA, Moroccan). While global model performance differs by only ~1%, stacked autoencoder models (especially Models 3 and 4) exhibit clear improvements in classification accuracy for compositionally sparse or climatically relevant particle types, underscoring the importance of per-class evaluation over aggregated metrics.**

### 3.3 The Subtleties of Feldspar Species

The classification of feldspar species presents unique challenges due to their overlapping mass spectral characteristics. Our analysis reveals that the mass peaks 16 ($O^-$), 24 ($C_2^-$), 26 ($CN^-$, $H_2C_2^-$), 43 ($HCN^-$/$AlO^-$), 60 ($SiO_2^-$), and 76 ($SiO_3^-$) m/z, appear among the top influential features across all models, as shown in Figure 8. These peaks are universally discriminative, appearing in 44% of all "top 10 important feature" selections across models and particle types. This means that the same spectral features critical for distinguishing feldspars from other aerosols are also important for classifying entirely different particle types, creating systematic overlap that impacts classification accuracy, particularly between K-feldspar and Na-feldspar.

Across all models, Na-feldspar proves consistently challenging to classify, showing the lowest F1 scores among feldspar species. Even Model 1's best performance (F1 score: 75%) exhibits significant misclassification rates, with 7% incorrectly classified as K-feldspar and 1 out of 10 samples as non-feldspar species. K-feldspar shows better classification performance but demonstrates potential classification bias, as **other feldspar** species are more likely to be misidentified as K-feldspar (>7.3%) than Na-feldspar (<3.4%). Models 3 and 4 demonstrate better performance in distinguishing between feldspar species (AUC-PR scores >0.93) compared to SVM-based models (<0.90), suggesting the autoencoder approaches better capture subtle spectral differences.



Feldspar cSA and cSOA show distinct classification patterns from their uncoated counterparts. Model 4 achieves 95% precision for Feldspar cSA with only <2% false positives relative to other Feldspar species. These coated variants share

fewer common mass peaks than the uncoated Feldspar species (only m/z 1 (H$^-$), 25 (C$_2$H$^-$), 26 (CN$^-$, C$_2$H$_2^-$), and 43 (C$_2$H$_3$O$^-$)), enabling better discrimination between species. Model 3 achieves the highest F1 scores for both coated species (88% for cSA, 87% for cSOA). The presence of more unique influential mass peaks, including m/z 19 (F$^-$), 50 (H$_2$SO$^-$), 79 (PO$_3^-$), 85, 87, 101, and 115, aids in distinguishing these species, contributing to their more reliable classification.

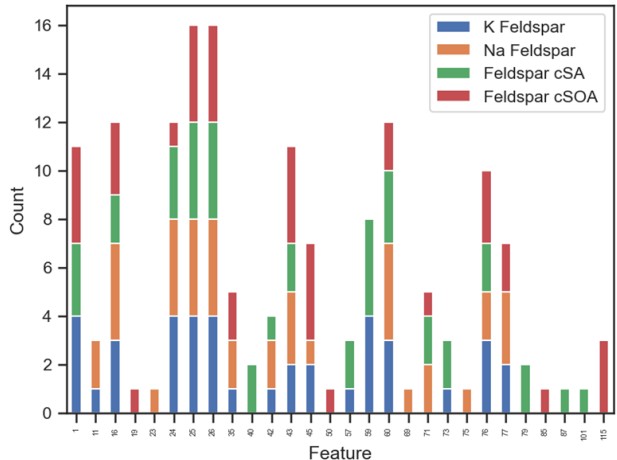

**Figure 8. Top 10 Feature importance counts for Feldspar species across all four models.**

When the four feldspar classes in the confusion matrix are merged into a single class, and the performance metrics were recalculated, it became evident that models 3 and 4 outperformed models 1 and 2 in differentiating Feldspar species from other aerosols. Model 4's performance increased overall accuracy from 90.6% to **94.4%**. This approach achieves 96% precision and 94% recall on the test set, though misclassifications persist, primarily with Chinese, ATD, Illite, and Ethiopian

samples. This performance improvement suggests that while distinguishing **between feldspar species remains challenging**, the models effectively identify feldspars as a broader class.

**3.4 Classification Performance by Aerosol Type**

Correct identification of organic aerosols such as agar, bacteria, hazelnut pollen, snomax consistently yields high F1 scores across all models, along with classification of Fly ash. Organic aerosols show high recall rates of 98-99%, suggesting their

chemical signatures are distinctive and well-captured by the classification algorithms. Notably, Bacteria and Hazelnut achieve perfect recall (100%) across all models, while Snomax maintains perfect precision. The strong performance in classifying Fly ash can be attributed to its substantial representation (21%) in the labeled dataset and its diverse characteristic peaks (1, 16, 24-26, 43, 76, 77, 79, and 80 m/z) that align with overall important identifying features across models.





The particle time-of-flight as proxy for the aerodynamic diameter emerges as a crucial distinguishing feature, particularly in SVM-based models (Models 1 and 2), where it appears among the top 30 features 15 times, compared to only twice in autoencoder-based models (Models 3 and 4). This parameter proves especially valuable for differentiating between mineral aerosols (8 instances) and organic aerosols (9 instances). The feature's significance is particularly evident in classifying Bacteria and Snomax, where size distributions, as seen in Fig 8, show distinct patterns: Bacteria exhibiting larger sizes while Snomax particles form a well-defined unimodal distribution.

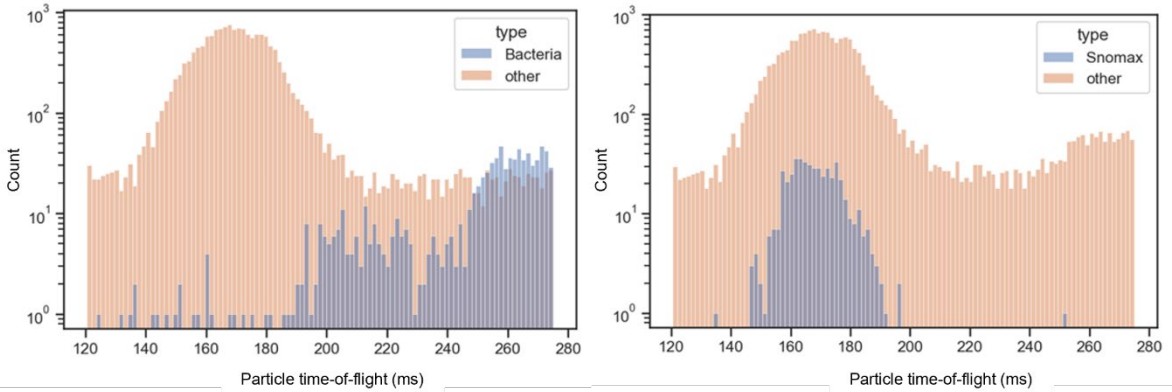


**Figure 9. Size distribution of Bacteria and Snomax classes vs the rest of the samples in the labeled dataset.**

Remarkably, soot and hazelnut pollen achieve F1 scores above 93% and often reach 100% despite their limited representation in the dataset. Hazelnut pollen maintains perfect recall across all models, while soot's recall ranges from 93% to 100%. The stacked autoencoder models (3 and 4) successfully classify all soot samples, whereas models 1 and 2

misclassify a single soot outlier as fly ash, likely due to sparse spectral features at 24 ($C_2^-$), 36, and 49 m/z that are better captured by the autoencoder architecture. The F1-scores of these classes are shown in Table 3.

| Category | Model 1 | Model 2 | Model 3 | Model 4 |
|---|---|---|---|---|
| Soot | 0.93 | 0.93 | **0.97** | **0.97** |
| Hazelnut pollen | **1.00** | **1.00** | **1.00** | 0.97 |
| Agar | **1.00** | **1.00** | 0.96 | **1.00** |
| SOA | **1.00** | 0.98 | 0.96 | 0.98 |
| Moroccan soil dust | 0.92 | 0.91 | 0.93 | **0.94** |

**Table 3. Model Performance for Lower Sample Size Classes.**

The models show effective performance in distinguishing geographically distinct dust aerosols (Moroccan, Ethiopian, and Argentinian soil dust), despite their smaller sample sizes and overlapping spectral features, particularly in the 25-26, 43, 64,

76, and 79 m/z ranges. Ethiopian aerosols are characterized by significant features in higher m/z ranges, while Argentinian soil dust show distinctive features below 100 m/z. Moroccan soil dust aerosols display higher overall spectral activity,





though some soil samples are occasionally misclassified as Ethiopian or Argentinian due to shared spectral characteristics. Despite these challenges, the models maintain robust recall rates across these geographic variants.

## 4. Discussion

This study demonstrates the potential of machine learning approaches, particularly stacked autoencoders, in advancing the classification of atmospheric aerosols using single-particle mass spectrometry data. The overall accuracy of 90-91% achieved by our models is competitive and represents an advance for fine-grained aerosol analysis. Previous work on the same 20-class dataset using a Random Forest classifier reported an accuracy of 87% (Christopoulos et al., 2018). Our results surpass this benchmark, indicating an improvement in classification reliability. Furthermore, our performance is on par with

state-of-the-art deep learning methods, such as 1D and 2D CNNs, which have achieved accuracies of approximately 90.4% on similar, albeit less complex, 13-class SPMS datasets (Wang et al., 2023, 2024b). While higher accuracies (94-97%) were reported, these are for a broader 8-class problem (Wang et al., 2024c), underscoring the strong performance of our models on this challenging 20-class task.

The stacked autoencoder classifiers (Models 3 and 4) showed a slight increase in performance compared to traditional SVM-

based approaches (Models 1 and 2). To paint a better picture, we diagnose the reconstruction and performance per class capabilities of these models, as well as identify potential sources of confusion.

The reconstruction quality of the stacked autoencoder varied across aerosol populations, this variation showed no direct correlation with classification accuracy. This suggests that the model's primary strength lies not in its reconstruction capabilities but in its effective encoding and feature extraction within the compressed latent space (right panel of Fig. 4),

allowing it to capture subtle spectral patterns that distinguish different aerosol types.

Our analysis revealed patterns in how different model architectures approach classification. Aerodynamic diameter emerged as a crucial feature, particularly for SVM models classifying Snomax and Bacteria, reflecting their distinct size distributions. In contrast, autoencoder-based models demonstrated the ability to leverage a broader spectrum of features while maintaining sensitivity to size-related characteristics. This difference in feature utilization highlights the nature of different modeling

approaches and suggests potential benefits in developing ensemble methods that combine their strengths.

The models demonstrate robustness in handling imbalanced training data, achieving high performance in classifying aerosols with low representation in the dataset, such as Soot and Hazelnut. At the same time, significant challenges persist in classifying certain species, particularly feldspars, due to overlapping spectral features.

The systematic difficulty in distinguishing K-feldspar from Na-feldspar stems from multiple factors: sample size disparities,

potential unidentified outliers, and overlapping classification boundaries due to shared influential features, particularly evident in the correlation between mass peaks 25 and 71 m/z (r = 0.94). Conversely, the higher performance in classifying coated feldspars (cSA and cSOA) suggests a higher sensitivity to surface composition, which has implications for understanding aerosol mixing states. Overall accuracy of model 4 is increased by 4% when aggregating the four feldspar



species into one class. Distinguishing between feldspar species remains challenging but separating them from the rest of the classes proves an easier task.

The integration of unlabeled data through semi-supervised learning techniques yielded mixed results. While the self-learning SVM showed limited improvement, the stacked autoencoder approach successfully leveraged unlabeled data to enhance classification accuracy. This finding underscores the importance of model architecture in effectively utilizing unlabeled data, a crucial consideration given the typical scarcity of labeled data in atmospheric science applications.

## 5. Conclusion

This paper develops and evaluates machine learning approaches for classifying atmospheric aerosols using single-particle mass spectrometry data, with particular focus on leveraging unlabeled data through semi-supervised learning techniques. We compared four models: supervised SVM, self-training SVM, stacked autoencoder, and mean teacher autoencoder frameworks on a dataset of 18,827 labeled spectra representing 20 distinct aerosol types. To demonstrate the performance of the deep learning approach, we showed that stacked autoencoder models achieved 91.1% overall accuracy, outperforming SVM-based methods (90.0-90.3%) while maintaining robust performance for rare aerosol classes critical to atmospheric science applications. The autoencoder architecture successfully leveraged unlabeled data during representation learning, whereas traditional semi-supervised SVM approaches showed limited improvement. These results advance automated aerosol classification capabilities and provide methodological insights for applying machine learning to atmospheric particle analysis, ultimately supporting more accurate characterization of aerosol impacts on climate and air quality.

Future work requires addressing three key limitations identified in this study. First, developing targeted approaches for class imbalance, particularly for rare but scientifically important aerosol types like Soot and biological particles where limited training data constraints model development. Second, improving discrimination between chemically similar species, especially feldspars, through advanced feature engineering or architectural modifications that better capture subtle spectral differences. Third, enhancing semi-supervised learning frameworks to more effectively leverage the abundant unlabeled SPMS data typically available in atmospheric measurements. Addressing these challenges would improve classification reliability for operationally critical but underrepresented aerosol populations and maximize the value of unlabeled spectral data in real-world deployment scenarios.

## Acknowledgments

The authors would like to acknowledge the essential role of the PyTorch and Scikit-Learn Python libraries in the development and implementation of the machine learning models utilized in this study. The authors would like to thank the KIT AIDA facility staff for hosting the FIN01 workshop and Daniel Czizo for helpful comments on this article.



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
