# Peer review of "Leveraging Machine Learning to Enhance Aerosol Classification using Single-Particle Mass Spectrometry"

_EGUsphere, 2025_

## Referee Comment (RC1)

**General Comments:**

Chavez et al. investigated two supervised learning methods (Support Vector Machine and stacked autoencoder classifier) and two semi-supervised approaches based on these two approaches for the classification of SPMS data. All four models achieved classification accuracies above 90% for 20 classes.

The topic fits within the scope of AMT, but I believe the manuscript does not yet meet the journal's quality standards. For example, the m/z values used for SPMS data classification are negative, but they were incorrectly presented as positive values in all of the text and figures. The description of the methods is unclear, making it very difficult to reproduce the work from the text alone. The results section lacks reasonable interpretation, and moreover, two of the figures and the table present identical content. If the paper is to be accepted, these issues must be resolved prior to publication.

**Specific Comments:**

**Section Abstract Introduction**

1. According to the classification results in Table 2, the SVM outperforms its semi-supervised learning (Self-Learning SVM) across all four metrics. Similarly, the Stacked Autoencoder outperforms its semi-supervised version (Mean Teacher Stacked Autoencoder classifier) in three out of four metrics. These two semi-supervised learning methods performed worse than the supervised methods. However, in Abstract, Introduction, Discussion, and Conclusion, the authors consistently emphasize the semi-supervised learning rather than presenting interpretations or comparisons based on the actual results. Similar issues in other sections also need to be revised accordingly.

2. Line 81

Add and cite recent research works.

3. Line 83

KMeans should be changed to K-means throughout the text.

**Section: Data and Methods**

1. Lines 147 to 150 state that each mass spectrum contains 193 mass-to-charge (m/z) peaks, so the feature range should be from -1 to -193. The manuscript instead reports a range of 1–207; this discrepancy must be resolved.

Furthermore, the m/z values should be negative (line 150), but in the text and figures present all m/z as positive (examples: lines 340, 355, 357, 373, 385, 389, 390, 421; Figures 4 and 8). This is a fundamental error.

2. The dataset contains 18,827 labeled samples divided into 20 classes. Due to class imbalance, the manuscript must include a table showing the number of samples and their proportion in each class. In addition, during preprocessing, you will drop the spectra with only one or fewer peaks. How many samples remained after this preprocessing step, and what was the distribution of samples across classes after preprocessing?

3. An 80% training / 20% testing split is widely used. How did you validate and decide to use 10% instead of 20% or 25% as mentioned in line 169? What are the results when using 10%, 20%, and 25% data for test?

Classes such as Soot, Hazelnut, and Agar each account for only about 1% in the dataset (line 154). With a 10% test split, these classes contain only a dozen or so samples, which makes the results highly random and unrepresentative. In theory, the classification accuracy of minority classes should be lower. However, as shown in Figure 6, the results indicate 100% accuracy for Hazelnut, and Aagar classes. This is most likely an artifact caused by the very small number of test samples.

4. Were the same labeled data used for training and testing across all four methods? The authors mention using 3-fold cross-validation to train the Self-Training SVM Classifier (line 206). However, cross-validation requires splitting the training set further into training and validation subsets, and the manuscript does not provide sufficient details. Were the other three methods also trained using cross-validation?

If the labeled training and testing data differed among the four methods, then the results are not comparable.

5. The unlabeled dataset includes 14,478 mass spectra. How many classes are represented within this unlabeled set? In Model 2, about 25% of unlabeled data were used. How many unlabeled data were used in Model 4?

6. Line 180

Use level 3 headings for the titles of the four methods, such as 2.1.1 Support Vector Machine Classifier, 2.1.2 Self-Training SVM Classifier.

7. Line 182

Citation error, remove (Christopoulos et al., 2018)

**Section: Results**

1. In the Results section, the presentation of the same metrics is very inconsistent. For example, in Table 2, values are reported as decimals, whereas in the text most are given as percentages (lines 311–320), but sometimes decimals are used again (line 330). Throughout the manuscript (text, figures, tables), metric values must be presented consistently—either all as decimals or all

as percentages. Additionally, there are two instances where figures and tables contain identical content, which is redundant and should be corrected. The content of Table 2 and Figure 5 is identical; Figure 5 should be removed. Similarly, the content of Table 3 and Figure 7 is identical; Figure 7 should be removed.

2. Line 340

How did you analyze the importance of the ions?

3. Line 346 - 358

Error analysis need some mass spectra as example.

4. Line 374, Line 411

Have you trained models separately with and without aerodynamic diameter and compared their results? The aerodynamic diameter accounts for only one feature out of 194, so its relative weight is just 1/194.

---

## Community Comment (CC2)

**Leveraging Machine Learning to Enhance Aerosol Classification using Single-Particle Mass Spectrometry**

https://doi.org/10.5194/egusphere-2025-3616

**General comments**

The work submitted for publication reports an interesting study on ways to improve the accuracy of classifying aerosol particles - which were ionized and analyzed by Single-Particle Mass Spectrometry - by means of Machine Learning. The proposed semi-supervised learning, in which unlabeled data is used for learning, is undoubtedly of great importance in practical applications.

To date, only a few approaches to semi-supervised learning (even beyond SPMS) are known and have been cited in the paper. The efforts undertaken in this study are very welcomed and promising. The chosen approach can be considered largely novel.

Nevertheless, a tailored implementation with convincing results and 'design guidelines' to achieve the best results would be of considerable significance for many applications.

The text is well written and very informative, with only little but disturbing redundancies. E.g. Table 2 and Fig. 5 as well as Table 3 and Fig. 7 bear the exact same information. It is recommended to omit Figs. 5 and 7.

**Major issues**

(1) The study's aim is to propose sophisticated Machine Learning models capable of bringing the classification performance closer to the optimum of 100 %. The obtained accuracies for the four described algorithms are surprisingly similar to each other (90.0% to 91.1%), with a significant gap to the optimum. This means, looking at the dataset as a whole, almost 10% of the assignments are incorrect. It is worth discussing how these incorrect assignments (false negatives and false positives) would be handled in practical applications.

From the results one might draw the conclusion, that systematic weaknesses common to the different approaches prevent better results from being achieved. The authors speculate on some of the causes (imbalanced dataset, number of classes, similarities between spectral features), but the dependence on these factors is not investigated.

- (2) It is suggested to take a closer look to one of the most prominent difficulties for Machine Learning models which is a heterogeneous, limited, imbalanced training dataset.
- (a) The dataset chosen by the Authors is very heterogeneous. It contains mass spectra of aerosol particles from very different emission sources, collected in various measurement campaigns. Part of the dataset (it remains unclear, what proportion) was used in a historical reference (Christopoulos et al., 2018).
- (b) The dataset is comparatively small (less than 20,000 labeled spectra), nevertheless comprising samples of as much as 20 (!) different classes of aerosol particles. Hence, on average, there are less than 1,000 labeled samples per class in the dataset. The test is performed on 10 % of the dataset, which for the under-represented classes (soot, pollen, agar) leaves less than 20 labeled test samples.
- (c) The class sizes vary greatly, from 21% to 0.8% of the total number of spectra. Such strong class imbalance is a well-known obstacle for high-performance ML applications. Methods to balance the class sizes via data augmentation are mentioned and cited in the text, but were not applied. Moreover, the

greatest advantage of semi-supervised learning and probably its core motivation is that the training dataset can be balanced and enlarged with almost no effort by adding unlabeled data to it. To exploit this advantage was apparently not considered by the Authors.

- 3) In the Introduction, the Authors criticize the common practice of assigning all samples in the dataset to a fixed number of predefined classes, without the option to classify certain samples as 'unknown'. In the presented implementation, however, such class comprising all samples of 'uncertain' or 'unknown' origin is still missing. The authors apparently quietly assume that all unlabeled mass spectra can be assigned to one of the 20 defined classes.
- 4) To improve the significance and practical applicability of the presented novel promising self-training and autoencoder classifiers, is it recommended to demonstrate their potential by a step-wise approach, starting from a sufficiently large, homogeneous, balanced dataset with only a few classes, to achieve a classification accuracy close to 100%. Then, step-by-step the dataset can be made more 'complicated' in various ways (increasing the share of unlabeled data in the first place), to draw implications for the usability of the sophisticated classifiers for various applications. Certainly, only few applications will need to classify unknown mass spectra into 20 very different classes like feldspar and agar.

**Minor issues**

- 1) In Lines 142-145, the dataset is defined as being composed of data collected during a FIN Workshop (reference from 2024) and data already used by (Christopoulos et al., 2018). In Lines 311-314, it is stated that the achieved overall accuracies of 90-91% surpass the 87% accuracy previously reported using the (Christopoulos et al., 2018) dataset. Apparently, the results are only comparable when the same data were used, hence when the data from the FIN Workshop were excluded. Was this the case?
- 2) Line 209ff and Figure 1: Are the numbers correct? In Line 158 it was given that 14,487 unlabeled mass spectra were included in the dataset. As can be read from Figure 1, for confidence threshold 0.95, about 12,000 labeled spectra were used (why not all?). How does this match to the 25 % of the unlabeled spectra incorporated in the training set? Does it mean that the fraction of unlabeled data in the training set is fixed and roughly 30 % (~4,000/~12,000)? Was the same dataset or the same proportion of unlabeled to labeled samples used for all algorithms?
- 3) Lines 231ff. How the parameter values (e.g. 96 latent dimensions, 48 features) were found?
- 4) Line 295: "Reconstruction quality shows no correlation with classification accuracy". That's a bold claim! It might be true only for the specific dataset.
- 5) Line 411: "Our analysis revealed patterns in how different model architectures approach classification." This sentence is difficult to understand. What are its consequences?
- 6) The subscript to Fig. 8 is cryptic. It should be explained, that for every of the 4 Feldspar species up to 4 score points can be gained for 4 models.

**Conclusion**

The work presents a valuable but rare approach to classify a mix of labeled and unlabeled data based on semi-supervised Machine Learning. Four algorithms and their classification results are presented in greater detail. For better understanding, the manuscript needs some clarifications and corrections. Recommendations are given how to re-design the study in order to improve the practical value and significance of the results.